# RSNN: Recurrent Spiking Neural Networks for Dynamic Spatial-Temporal Information Processing

## ABSTRACT

Spiking Neural Networks (SNNs) have great advantages in discrete event data processing because of their binary digital computation form. However, due to the limitation of the current structures of SNNs, the original event data needs to be preprocessed to reduce the time calculation steps and information redundancy. The traditional methods of dividing data into frames lead to the loss of a large amount of time information. In this paper, we proposed an efficient Recurrent Spiking Neural Network (RSNN) to reduce the time domain information loss of original slice samples with the spiking based neural dynamics for processing the dynamic spatial-temporal information. By constructing the Recurrent Spiking Neural Network model, the recurrent structure was used to preprocess slices before it was further input into the spiking structure to enhance the time correlation between slices. In addition, in order to match the two-dimensional spatial structure of data sample frames efficiently, this paper adapts a variation of structures of the recurrent neural network, named Convolution LSTM (CONLSTM). Through experiments on event based datasets such as DVS128-Gesture and CIFAR10-DVS, we find that the proposed model could not only behave better than some other spiking based models but also save energy and power consumption which paves the way for practical applications of neuromorphic hardware.

## KEYWORDS

Spiking Recurrent Neural Networks, Dynamic Spatial-Temporal Information, Event-driven, Neural Dynamics

## 1 INTRODUCTION

INSPIRED by the neural dynamics of biological neurons [3], Spiking Neural Networks (SNNs) were proposed to bridge the gap between biological neuroscience and artificial intelligence by using discrete spikes as communication carrier [8, 19, 23, 37]. This binary and discrete form of computation makes SNNs have lower computational power consumption and better anti-noise ability [17, 31]. Moreover, there are various advantages in the practical deployment of neuromorphic hardware because of the event-driven nature [1, 4, 18, 20], which makes SNNs regarded as the next generation of neural networks [22, 25]. Neurons of the SNNs use the form of spike signal integral-fire for data processing. The membrane potential on the spiking neurons is gradually accumulated under the influence of

*ACM MM, 2024, Melbourne, Australia*
© 2024 Copyright held by the owner/author(s). Publication rights licensed to ACM.
ACM ISBN 978-x-xxxx-xxxx-x/YY/MM
https://doi.org/10.1145/nnnnnnn.nnnnnnn

the spike signal transmitted by other neurons. When the membrane potential value is higher than the firing threshold, the spike will be fired outward. This mode simulates the dynamics of real biological neurons, which makes SNNs have powerful memory ability and are very suitable for processing dynamic spatial-temporal information. However, due to the discrete data form of SNNs, the value is not derivable at the point of spiking transmission when using the backpropagation (BP) [11] based method to train the parameters in SNNs.

Unlike static images, dynamic spatiotemporal patterns with neural spikes could express more information[33]. However, most current spiking based models could not leverage these strengths, among which network structure is an important one. Most of the structures of typical SNNs are shallow and fully connected [26, 28], which means these kinds of structures could not extract and encode external features adequately. Although some works tried to construct ANN-SNN mixed models [36, 38], they were proposed for static image recognition, they cannot be applied to dynamic spatio-temporal information recognition directly.

CSNN [35, 36], they drew inspiration from the visual pathways in the biological visual system that process static images. They constructed a network that blends Convolutional Neural Networks (CNNs) with Spiking Neural Networks . This network structure employs convolutional layers to capture spatial features from static images, thus enhancing the feature extraction capabilities of the Spiking Neural Networks, leading to improved performance. However, traditional two-dimensional convolutions are generally used for processing static images, limiting the use of this Convolutional Spiking Neural Network (CSNN) to static data. To address this limitation, we aimed to develop a feature extraction structure with spatiotemporal processing capabilities. We achieved this by combining the Convolutional LSTM structure with the Spiking Neural Network, resulting in the proposed Recurrent Spiking Neural Network (RSNN). RSNN is grounded in principles inspired by biological visual mechanisms, making it more versatile and no longer confined to static visual stimuli.

We use neuromorphic datasets to evaluate the performance of RSNN. As one of time series based spatio-temporal information, the neuromorphic vision datasets are acquired by Dynamic Vision Sensor (DVS) [32] in real time to collect the dynamic scene of the outside environment. So due to the spike based nature of both SNNs and DVS based datasets, it is a natural association to build a bridge between them. Some researchers tried to build a deep SNN [5, 6, 14, 15]to recognize images in DVS-CIFAR10 [39]. This work [27] constructed an SNN model to recognize scoring in basketball games with an address event representation (AER) sequence. Someone [21] wanted to utilize neural dynamics to extract the spatiotemporal information that is hidden in time series data. Inspired by human brain, this work [10] analyzed temporal and spatial characters of electroencephalographic signals.

While SNNs exhibit versatility in the tasks mentioned above, they have limitations in extracting performance from dynamic spatiotemporal information. Dynamic data contains abundant temporal information as well as intricate spatial details. SNNs effectively capture temporal information due to their time-based memory capabilities but often struggle to efficiently extract spatial texture features from each frame at each time step. This limitation contributes significantly to the restricted performance of Spiking Neural Networks. In our work with RSNN, the Convolutional LSTM plays a pivotal role in enhancing performance. It preserves temporal information while efficiently extracting spatial texture features through convolution. Additionally, the discrete pulse-triggered data computation pattern of SNNs grants RSNN a notable advantage in energy efficiency, which we have validated through experimentation. We evaluated the proposed method on several event-driven datasets (DVS-128Gesture [2] and CIFAR10-DVS [16]). The main contributions are as follows:

- This paper proposed an RNN-SNN hybrid model that adopted RNN for feature extraction and SNN for feature recognition. The proposed hybrid RSNN model can learn more sensory information by RNN based structure, and SNN part could classify the event based spatial-temporal information adequately.
- Through the proposed hybrid RSNN model, the feature extraction and encoding were ensembled into one framework. We design a particular RNN-SNN training method for training the proposed model by combining the advantages of BP and surrogate gradient methods respectively.
- We demonstrate the efficiency and effectiveness of the proposed RSNN model after evaluating it on two time series datasets. Experiments show that we can construct such a reasonable hybrid structure and it can achieve state-of-the-art performance on those event-driven datasets with less computational power with full use of neural dynamics.

## 2 METHODS

Inspired by the information processing mechanism in biological neural systems, we propose a hybrid network structure with RNNs as a feature extractor and Spiking Neural Networks as a decision maker. The proposed model is called Recurrent Spiking Neural Network (RSNN) as shown in Fig.1. The RNNs part of this proposed model framework structurally mimics the V1 of the retina in human brain which provides a strong reference for research on neural networks in the field of vision. The SNNs part as a classifier acquires the image features after spike encoding from the feature extractor and learns network parameters through the SNN training method. RSNN model is a unified system model that integrates feature extraction, coding, and learning.

### 2.1 Overview of the proposed RSNN model

In the processing of visual information by the biological neural systems, external visual stimuli are extracted by the visual cortex in a hierarchical structure, and these features are then fed to the cerebral cortex in the form of bio-spike electrical signals for subsequent decision processing. It is this efficient feature extraction model that allows the biological neural systems to perform well on visual tasks. The visual signal is generated by retina acquired from the external environment and then transmitted through the Lateral geniculate nucleus (LGNs) to the visual cortex, where the features of image signals are efficiently extracted by a multi-layered physiological construct. These layer-by-layer feature extraction operations capture features of original information from spatial and temporal perspectives. In previous work, the feature extraction function of the biological visual neural systems in spatial dimension was mainly simulated, which showed high accuracy and noise immunity in static image recognition tasks but was not effective for temporal data recognition due to the lack of feature extraction modules in the temporal dimension. Our proposed RSNN model focuses on feature extraction operations in spatiotemporal dimension of bio-vision systems by analogous and migration with RNN modules.

In RSNN, the ability of RNN to remember information makes the image signals in previous moments also affect the feature extraction at the current moment. Through the ability of temporal memory, the performance of feature extraction in temporal dimension belonging to RSNN is promoted to a higher level. For feature extraction in the spatial dimension, we add a convolution module to the basic RNN network framework. Before the formal cyclic processing of the data stream, we obtain feature extraction in spatial dimension through convolution and pooling operations by convolving the image frames of the current and past moments and reducing the image size by a pooling layer at each moment of output. These operations of the network are also biologically interpretable. The operation of the convolution layer simulates the extraction of important external optical information at biological retina, and CCs layer in the cerebral cortex is also biological basis for layer-by-layer extraction of image features by the convolution layer. Spatiotemporal dynamics mechanism of biological neurons, which enables the biological visual nervous system to have the ability to remember information in temporal dimension. The biggest innovation of our work is referring to this structural property, by introducing a convolutional RNN module, to exploit the recognition and classification ability of the RSNN system model for neuromorphic temporal datasets within the scope of biological interpretability. In addition, considering the attention mechanism of the human visual system for complex dynamic images, we also add a self-attention component to RSNN to further improve overall performance.

### 2.2 Feature extraction and encoding

The neuromorphic datasets used in our work are collected by DVS camera. Through the camera's sensitive unit, it senses changes in the intensity of the external light, resulting in a spike signal. Because the imaging mode of DVS camera is to feel the change of light intensity and only respond to the change of external lighting conditions to generate spike events[24], it has the ability to image moving objects. Compared with the conventional camera in the accumulation of light imaging within the exposure time, DVS camera can more accurately present high-speed moving objects, which is not easy to cause motion blur and other problems. And, the output data form is discrete binary spikes, which is consistent with the input data form of SNN and is also convenient for further data processing in the later period. This neuromorphic event data can

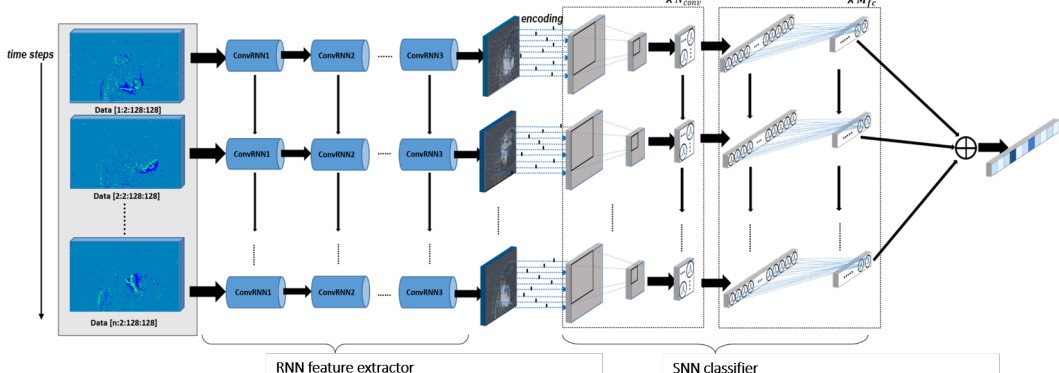

**Figure 1: RSNN model**

be expressed as $E(x_i, y_i, t_i, p_i)(i = 0, 1...N - 1)$ Where $(x_i, y_i)$ represents the coordinate position within the horizon of the camera lens where the event is located, $t_i$ represents the moment when the event occurs, and $p_i$ represents the channel where the event is located. Neuromorphic event data has two channels, which respectively represent the spikes generated by the enhancement of light and the spikes generated by the attenuation of light. For DVS cameras, the changes in the direction of external illumination enhancement and attenuation will produce events, and the corresponding $p_i$ values are -1 and 1, respectively. The DVS camera generates an event in a few microseconds on average, and the amount of event data generated in the sampling period of tens of milliseconds is very large. Therefore, it is necessary to preprocess the dataset first. We use slicing as a data preprocessing method, and the specific approach will be explained in the experimental section.

Considering that the conventional RNN model is prone to problems such as gradient explosion and gradient disappearance when processing long time series data, as a result, it is difficult to process data samples that are deep in time. In response to it, we use a variant of RNN, Long Short Term Memory (LSTM) network in this paper. By adding some gating mechanism, this variant model can retain the past information, which is able to restrain the long-range dependence problem to a certain extent effectively. The gating mechanism of LSTM is realized through a continuously guided gating function. There are three kinds of gates in this network, they are input gate, forget gate and output gate. The input gate is used to control the retention degree of neuron state value at the current moment, the forget gate is used to control how much neuron state at the previous moment needs to be forgotten, and the output gate controls how much neuron state at the current moment outputs to the hidden layer state of the neuron. In addition, different from the conventional discrete binary gating function, the gating unit in the LSTM model has not only 0 and 1, but values between them, indicating that a certain proportion of information is allowed to pass through. It is more common for the gating function to adopt a logistic function. Considering that the form of data to be processed is a two-dimensional image, in order to better fit the structure of data and facilitate feature extraction, we add convolution operation on the basis of LSTM structure to construct Convolutional LSTM

(CONVLSTM).The Convolutional LSTM network was initially introduced in a paper[29], which blends the concepts of CNN and RNN. This integration enables the network to extract both temporal and spatial information effectively. Extensive research has shown that CONVLSTM networks perform well in tasks involving video frame sequences. The expression of the network is:

$$i_t = \sigma(W_{xi} * X_t + W_{hi} * H_{t-1} + b_i) \tag{1}$$

$$f_t = \sigma(W_{xf} * X_t + W_{hi} * H_{t-1} + b_f) \tag{2}$$

$$o_t = \sigma(W_{xo} * X_t + W_{hi} * H_{t-1} + b_o) \tag{3}$$

$$c_t = f_t \circ c_{t-1} + i_t \circ tanh(W_{xc} * X_t + W_{hc} * H_{t-1}) \tag{4}$$

$$H_t = o_t \circ tanh(c_t) \tag{5}$$

Where, the symbol $*$ represents the convolution operation, and $\circ$ is the haddam operator. The variable $c_t$ stores the information about the model unit, which is controlled by both the input gate and the forget gate, and then the variable is controlled by the output gate after input to the $tanh$ function to obtain the hidden value $H_t$ of the model hidden layer at the current time. Hidden values preserve the state information accumulated by the model at the last moment and guide the output at the next moment with the input. In the equation, the CONVLSTM model is different from the one-dimensional LSTM, which convolves the input at the current time with the hidden state at the previous time. Using two-dimensional convolution to carry out data input at each fixed moment, the convolution of the static frame image at this moment can better extract the features of the static image frame. In addition, with the convolution of hidden layers of the past time, the CONVLSTM model can have a more accurate grasp of the sample features on both the dynamic time sequence and the static image by embedding convolution operation in the LSTM sequential processing system.

In the course of processing external dynamic images, the biological visual system will pay more attention to some detailed features with more information, that is, spend more calculation resources on these features, which is also called the attention mechanism of biological vision. In order to achieve a similar function in computer vision, the attention mechanism is inspired by this. Our work also uses the self-attention mechanism to verify the ability of the model to extract dynamic features. We add the attention operation before each frame of the image is entered into the CONVLSTM. That

is, more important features are given greater weight through self-attention, and then further time sequence operations are carried out. The algorithm belonging to this kind of self-attention is introduced as follows. For a flattened image tensor $X \in R^{HW \times C}$, we define a query:$Q = XW_q$,a key:$K = XW_k$, and a value:$V = XW_v$.where $W_q, W_k \in R^{C \times d_k}$ and $W_v \in R^{C \times d_v}$ are learnable matrices.As shown in equation 6, self-attention is calculated by the dot product of tensor Q and K, and then the weight distribution coefficients of each feature are obtained by the softmax function. This coefficient is then multiplied by the eigenvalue correlation tensor V, thus giving more weight to important information and reducing the allocation of attention to non-important information. Since the attention coefficient is determined by both image X and the trainable weight, it is a kind of self-attention.

$$A_h(Q, K, V) = softmax(\frac{QK^T}{\sqrt{d_k^h}})V \qquad (6)$$

## 2.3 Spiking based classification

We use Spiking Neural Network as the decision module of the whole model. Spike neurons are the basic elements of SNN. Relevant researchers have established a variety of models based on the neuron dynamics of real biological neurons, such as Hodgkin Huxley model, which more accurately presents the ion flow process of biological neurons. Although the model structure of Leaky Integrate and Fired (LIF) neurons is simple, it also dynamically presents the changes of cell membrane potential, and the calculation is simple, which is convenient for the construction of large-scale networks. Our work uses LIF neurons. The complex intermembrane ion flow is ignored in the LIF neuron model, and only the process of macro membrane potential change is modeled mathematically. The expression is as follows:

$$\tau \frac{dV(t)}{dt} = -(V(t) - V_{rest}) + X(t) \qquad (7)$$

Where $V(t)$ represents the membrane potential at the current moment, $X(t)$ represents the stimulus received by the neuron at this moment, $\tau$ is a time constant, and $V_{rest}$ is the resting potential. The connotation expressed by the formula is that the change of the membrane potential at a certain time is determined by the current state of the membrane potential and the external input. There's also the concept of a membrane potential threshold $V_{th}$. When the cell membrane potential exceeds the threshold, the spike is emitted, and after that, the membrane potential is reset back to the resting potential $V_{rest}$. LIF reduces computational costs with uncomplicated modeling while retaining a certain amount of biological rationality. In order to facilitate the realization of this process by a computer program, the discrete expression form is given here:

$$H(t) = f(V(t-1), X(t)) \qquad (8)$$
$$S(t) = g(H(t) - V_{th}) \qquad (9)$$
$$V(t) = H(t) \cdot (1 - S(t)) + V_{rest} \cdot S(t) \qquad (10)$$

Where, $g$: a gated function and represents the spike emission process.$f$: Function that describes the network's state changes over time. $H(t)$: Represents the membrane potential of spiking neurons at the current time step. $S(t)$: Represents the spikes emitted by the spiking neural network at the current time step. $V_{rest}$: Represents the resting membrane potential of the spiking neuron.

As the trigger of the spike in SNN is represented by a gate function that cannot be differentiated, it is also difficult to apply the conventional BP algorithm to the parameter training of SNN. At present, the commonly used method is to convert ANN to SNN. This method is to directly assign the parameters trained by ANN directly to SNN. However, this method has two defects. First, the delay is large. In addition, since the weight training is completed on ANN and the parameters are not continued to be trained in the spike activity, it also loses a lot of spatio-temporal information. Another method for training SNN is gradient substitution, that is, the function with a higher slope replaces the non-differentiable gate function in the backpropagation of error so that it can be trained more conveniently by BP algorithm.

Your work should use standard LATEX sectioning commands: `section`, `subsection`, `subsubsection`, and `paragraph`. They should be numbered; do not remove the numbering from the commands.

Simulating a sectioning command by setting the first word or words of a paragraph in boldface or italicized text is **not allowed.**

## 2.4 Experimental settings

The experiments are evaluated on a server equipped with 5 cores Intel(R) Xeon(R) Gold 5218 CPU with 2.40GHz and 1 NVidia GeForce RTX 4090 GPU. The operating system is Ubuntu 20.04. Besides, we use spikingjelly [7] as our basic software simulation platform.

We experiment with the performance of our model on two datasets, DVS-128Gesture and CIFAR-10DVS. Among them, the DVS-128Gesture was obtained by sampling the presenter's action in real time with the DVS camera, and the CIFAR-10DVS was obtained by dynamically moving the static dataset in front of the DVS camera. Our RSNN model adopts the method of phased training, that is, the RNN module is trained first, and then combined into the RSNN network to train the subsequent SNN parts.

Our experiments are carried out on pytorch framework and spikingjelly, which is a framework for spiking neural networks developed on the basis of pytorch. We compared several RNN module structures to find the difference in their feature extraction capabilities. The differences of various methods will be introduced later. The network structure of RNN module we adopted is the structure of convolutional LSTM. The structure of SNN module is quoted in the paper[9]. This SNN is composed of several convolution layers and two fully connected layers. For different datasets, the number of convolution layers is different. For all convolution layers, the size of convolution kernel is 3, the stride is 1, and the padding is 1. We used LIF neuron as the basic element of SNN, and the time constant of the neuron was set as $\tau$=2.0. In the pre-processing of video samples, we adopt the method of average segmentation and addition, in which the number of frames of segmentation is T=20.

We utilized the data preprocessing approach provided by spikingjelly. The specific method was to select the time step T required for the experiment, and then each sample was evenly divided into T parts in the time dimension. We combine all the time information in a period of time into one frame. In the cut time segment, the events in the same spatial position are added together, and finally, the corresponding position of the frame is the addition result. This preprocessing method reduces the complexity of the original data

and provides convenience for subsequent data processing.In our experiments, we used a slice size of 20 frames.

## 2.5 RNN module pre-training

RNN module is responsible for the intensive extraction of dynamic time sequence features in the whole RSNN model. The main framework of the network is the CONVLSTM network. On this basis, some details are optimized and improved. Firstly, in terms of the module structure of RNN, we generally adopt two different structures, namely multi-layer CONVLSTM and convolution LSTM with self-attention mechanism. The purpose of conducting experiments on RNN of these two structures is to verify the advantages of self-attention mechanism in feature extraction and energy consumption. In addition, we found that, although the pooling layer can accelerate the training speed, it also reduces the scale of the picture, which affects the accuracy of the model. Therefore, the number of pooling layers and the training speed should be balanced. In addition, for the output of the CONVLSTM, we employed five different decision layer structures as follows:

**Take the last frame:** Take the output of the last time step of the convolution LSTM and input it into Convolutional Neural Network.

**Add all frames:** Add the frames output by each time step of convolution LSTM to one frame, then input to Convolutional Neural Network.

**All frames are averaged:**After averaging the values of the frames added together,then input to Convolutional Neural Network.

**Temporal dimension concatenation:** Concatenate frames at each time step along the temporal dimension, then input to Convolutional Neural Network.

**One-dimensional LSTM as the decision layer:** The output of the CONVLSTM, flattened over time steps, and then fed into a one-dimensional LSTM.

In the first method, since CONVLSTM has the memory ability in the temporal dimension, the content of the last output frame selectively remembers the content of the previous moment. Therefore, we believe that the last frame contains all the information of the moment to some extent, so we choose the information of the last frame as the subsequent input. In the second method, we add the values of frame output on all-time steps to obtain a new frame after summing. The reason for this operation is that the information of each time step can be well preserved because the selective memory of convolutional neural network cannot ensure that the useful information can be completely retained. In the third method, in order to simplify the data, we choose to average the values obtained by the second method. In the fourth method, the output vector shape of CONVLSTM is $(w, h, c, t)$, where $w$ and $h$ are the length value of the vector,$c$ is the channel numbers of output by the convolution layer, and $t$ is all the output steps of CONVLSTM. In order to retain the temporal and spatial information as comprehensively as possible, we hope to reduce the loss of information through as few simple processing operations as possible. So we superpose the features in the time dimension. In this way, the information in the time dimension is transferred to the channels, and all information is preserved comprehensively. Finally, the shape of the ground vector is $(w, h, c \times t)$. In the fifth method, in order to retain more information, we hope to use one-dimensional LSTM layer to replace the

original convolutional layer. We add the output of the LSTM of the last layer by step size to get the final output, and then judge the voting layer.

We visualized the output of the final layer of the CONVLSTM for these five methods, and the visualized results are shown in Fig.2. We present the feature diagram of each time step for each method. However, due to the difference in the number of frames processed by each method, we took out the CONVLSTM output of the trained RNN model and added all channels together at each time step. To make the feature representation more obvious, we enhanced each image. Each group of pictures has two lines, and each line has a total of 10 pictures, from the first picture to the 20th are the times step of the feature maps. In Fig.2, *origin img* is the sample of input, which is the temporal image of the action left hand in the DVS-128Gesture dataset. *AVG* is the feature image output by convolution LSTM after the training of the frame averaging method. It can be seen from the figure that the feature image obtained by this method can basically obtain the arc track drawn by the human arm on the left front of the human body, and as time goes by, the frames of subsequent moments retain part of the features of the frames at previous moments, that is, give play to the ability of timing memory. It preserves the characteristics of the present moment and the moments before it. However, it is also obvious that the noise of this method is quite serious, so that the human body is completely submerged in the noise, and the track of hand movement is not clear. The feature map obtained by *SUM* method is relatively clear, without so many chaotic noise particles as the previous method. However, the defect is that the contour of the arm rotation in the feature map at different moments is similar, and the position of the arm at the current moment is not highlighted, although remembering previous information, the realization of features is not particularly significant for left hand clockwise, which is a cyclical movement. The *lastframe* method does the same thing, focusing only on the broad outline of the arm rotation and not on any particular point in time. The *120channels* method, which produces a feature graph that lightens the arm rotation while highlighting the human body, we believe that this method is advantageous to sample classification because there is a right-hand clockwise category in DVS-128Gesture, The arm rotates on the right side of the human body, and the clear outline of the human body as a reference is more conducive to the differentiation of these categories. Finally, The *FC to LSTM* method. The feature map obtained by this method is the clearest among these methods, and the rotation outline of the arm and the human body is the most obvious. At different times, the outline of the arm does not look like a circle, which can be distinguished by the shades of color.

We analyze the performance comparison of two structures and five processing methods on two datasets. The two structures are multi-layer convolutional LSTM(CL) and attention-adding convolutional LSTM(CLA) respectively. The structures of the two models are shown in Table 1, where nC3 refers to a convolutional layer with n input channels and a convolution kernel size of 3×3. MPn refers to the max pooling layer with the pooling kernel size of n×n, and likewise APn refers to the average pooling layer. CLn refers to the CONVLSTM whose convolution kernel is n×n and the number of input channels is m, L and FC are one-dimensional LSTM and

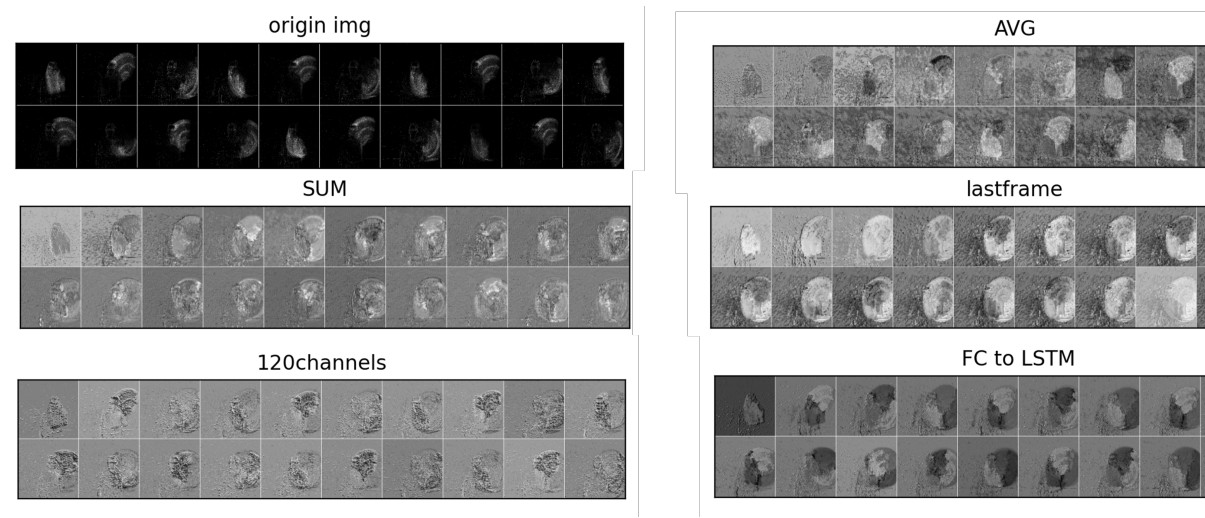

**Figure 2: output feature map comparison**

**Table 1: RNN module structure**

| Dataset | Model structure |
|---|---|
| **CIFAR10DVS** | CL(fc to lstm): Input-2CL3-MP2 -4CL3-MP2-6144L-1536L-384L-100-AP10

CL(others): Input-2CL3-MP2-4CL3 -MP2-6C3-MP2-64C3-64C3-MP2 -32C3-2048FC-512FC-100-AP10

CLA(fc to lstm): Input-AP4-2CLA3 -10240L-2560L-640L-100-AP10

CLA(others): Input-AP4-2CLA3-10C3 -MP2-64C3-64C3-MP2-32C3 -2048FC-512FC-100-AP10 |
| **DVS128Gesture** | CL(fc to lstm): Input-2CL3-MP2 -4CL3-MP2-6144L-1536L-384L-110-AP10

CL(others): Input-2CL3-MP2-4CL3 -MP2-6C3-MP2-64C3-MP2-64C3-MP2 512FC-128FC-110-AP10

CLA(fc to lstm): Input-AP4-2CLA3 -10240L-2560L-640L-110-AP10

CLA(others): Input-AP4-2CLA3-10C3 -MP2-64C3-MP2-64C3-MP2 -512FC-128FC-110-AP10 |

fully connected layers respectively. In addition, CLA represents the CONVLSTM with attention.

The main structure of the model is divided into two parts. The first part acts as the RNN module of the feature extraction layer.

The multi-layer CONVLSTM consists of two hidden layers, and the number of channels of the two hidden layers is 4 and 6 respectively. The CONVLSTM with added attention has a 4×4 average pooling layer and a convolution attention LSTM with one hidden layer. The second part for CNN or LSTM to make decisions through features, in the use of lastframe, sum, avg, 120 channels as the intermediate data processing method, with the CNN network as a decision module, for CIFAR-10DVS dataset, We used a four-layer convolution, and DVS128Gesture three-layer convolution. With the FC to LSTM method, we use a three-layer LSTM as the decision module. In order to facilitate the comparison of various methods, we plot the changes in the accuracy of several methods of different datasets on the same line chart which are shown in Fig.3. there are four graphs, where the first line is the curve of the multilayer convolution model and the second line is the curve of the model with added attention. The five methods are plotted in the same diagram. Since the size of the CIFAR-10DVS dataset is larger, we trained a total of 150 epochs, and for the smaller DVS128Gesture dataset, we trained a total of 300 epochs. We can get a general rule from the four figures, that is, the three basic methods, last frame, sum, and mean, their performance is not different, and the accuracy curve has a similar trend. However, although the method using LSTM instead of the full connection layer has the least intermediate information processing steps, it has the worst performance, because of the low complexity of the model. In other methods, the output of the previous RNN module is followed by a structure with multi-layer convolution, while the fc to lstm method directly uses three-layer LSTM to replace CNN, and its performance naturally decreases as the complexity of the model decreases. In Table 2 we show the optimal performance of each method.

## 2.6 RSNN model analysis

After the best RNN model is obtained, the combination of RNN and SNN can be carried out. We first chose the RNN module model with the best performance to combine with the SNN network. For the two

**Table 2: RNN module accuracy**

| model | method | DVS128Gesture | CIFAR10DVS |
|---|---|---|---|
| **Convolution LSTM** | lastframe | 0.944 | 0.684 |
| | sum | 0.958 | 0.695 |
| | mean | 0.955 | 0.678 |
| | 120channels | 0.861 | 0.642 |
| | fc to lstm | 0.771 | 0.399 |
| **Convolution LSTM with attention** | lastframe | 0.944 | 0.649 |
| | sum | 0.951 | 0.647 |
| | mean | 0.965 | 0.663 |
| | 200channels | 0.958 | 0.637 |
| | fc to lstm | 0.823 | 0.466 |

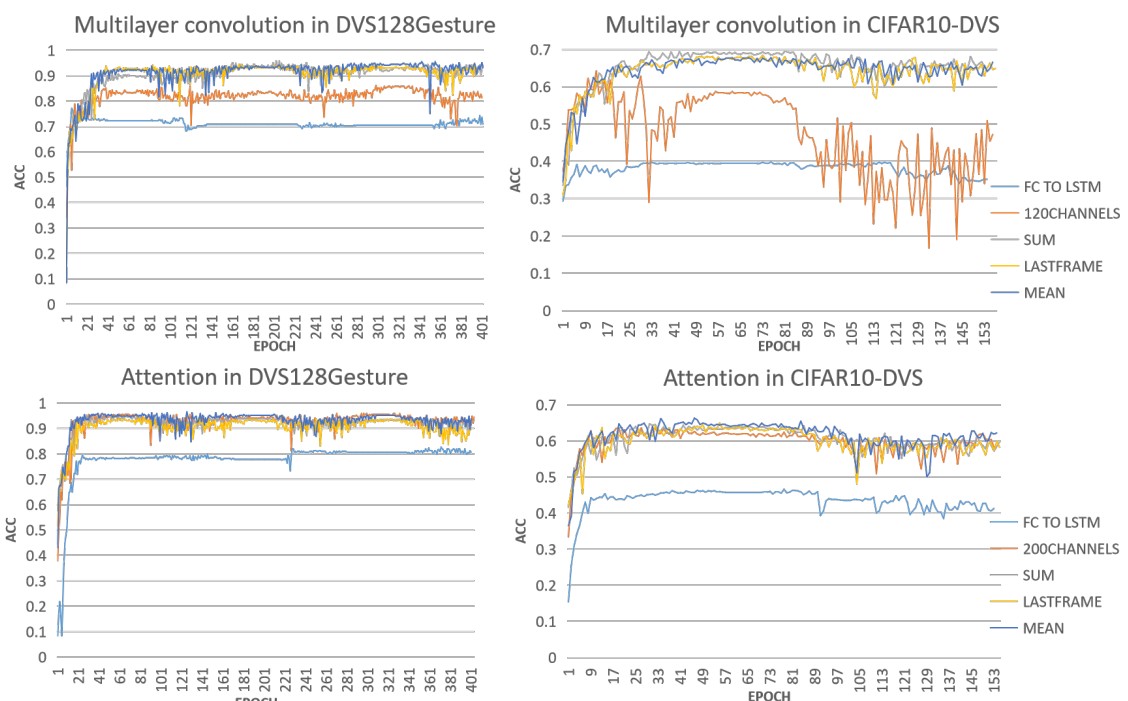

**Figure 3: Accuracy comparison**

**Table 3: The performance comparison between the proposed RSNN and other SNN models**

| model | DVS128Gesture | CIFAR10-DVS |
|---|---|---|
| Our RSNN(CL) | 0.955 | 0.669 |
| Our RSNN(CLA) | 0.951 | 0.671 |
| SCNN (4layer) | —- | 0.692 |
| SCNN (3layer) | 0.906 | —- |
| CNN-based SNN[12] | 0.936 | —- |
| SLAYER[30] | 0.934 | —- |
| STBP[12] | 0.934 | —- |
| STBP[34] | —- | 0.605 |

datasets, we adopt the SNN network in the form of two-dimensional convolution of different sizes. For CIFAR-10DVS, we adopt four-layer convolution SNN, and for DVS-128Gesture, we adopt three-layer convolution SNN. Here we call SNN of this convolution form SCNN, The structure of the two SCNNs is {128SC3-BN128-MP2-128SC3-BN128-MP2-128SC3-BN128-MP2-128SC3-BN128-MP2-128SC3-MP2-(128×m×m)SL-(32×m×m)SL-N×10} and {128SC3-BN128-MP2-128SC3-BN128-128SC3-BN128-MP2-128SC3-BN128-(128×m×m)SL-(32×m×m)SL-N×10}. Same as the expression before, SC here represents the convolutional layer of SCNN and SL represents the fully connecting layer composed of spiking neurons. In particular, m here is determined by the size of the image input. N refers to the number of categories, and its values are 11 and 10, respectively.

**Table 4: Energy Consumption Comparison between CONVLSTM(RNN) and RSNN**

|  | RSNN(CLA) | RSNN(CL) | CONVLSTM(RNN) |
|---|---|---|---|
| synops(1e4) | 2.31 | 1.33 | — |
| floats(1e4) | — | — | 27.67 |
| energy(1e-7J) | 0.21 | 0.12 | 12.73 |
| accuracy | 0.948 | 0.952 | 0.958 |

In order to verify that our RNN module can improve the performance of the entire RSNN network through its feature extraction capability, we conducted experiments on SCNN and RSNN. In the experiments of RNN modules, the CONVLSTM structure obtained the best performance on the two datasets when *SUM* method was adopted, and the CONVLSTM structure with attention was the best performance when *MEAN* method was adopted. Therefore, we adopt the RNN modules constructed by these two methods to form the RSNN network for subsequent experiments. The experimental results are shown in Table 3.

As shown in Table 3, the performance of the two RSNN networks is better than the SNN network in the DVS-128Gesture dataset, which shows that the addition of the RNN module improves the performance of the original model, because the structure of Convolutional LSTM promotes the further extraction of spatiotemporal details. In the comparison between the two RSNN networks, the performance of RSNN(CL) is slightly better. We analyze that there is a 4×4 average pooling layer in the first layer of the network model of CLA. This pooling layer is added in consideration of the limitation of hardware performance because it reduces the original input feature graph of 128×128 to 32×32. This also causes part of the spatial information lost, affecting the final performance. Therefore, RSNN (CLA) can maintain a similar performance to RSNN (CL) when the size of the input feature map is smaller, which also proves that the self-attention mechanism can promote the model to extract spatio-temporal information. However, the accuracy of RSNN model on CIFAR-10DVS is inferior to SNN model. We believe that this is because the pooling layer in RNN module reduces the size of the picture. The input image of SNN decision in RSNN is 32×32 and SNN is 128×128. So we retrain with the convolution LSTM without the pooling layers The result obtained is 0.711, which also proves the validity of RSNN on CIFAR-10DVS.

We compared the performance of our model with other models in DVS-128Gesture, and listed two models in total, SNN (CNN based) and SLAYER. The accuracy of our model was about 2% higher in the DVS-128Gesture, and this was based on the premise that there were fewer SNN layers, and our model had more advantages in power consumption and performance.

## 2.7 Energy consumption analysis

In the previous section, we mentioned that spiking neural networks have a significant energy consumption advantage over traditional artificial neural networks (ANNs), which we have also demonstrated in our experiments. For traditional ANNs, energy consumption is typically calculated based on the number of floating-point operations (FLOPs). According to the data provided in the paper[13],

using 32-bit floating-point implementation on a 45nm technology, $E_{MAC}$ (Energy for Multiply-Accumulate) is 4.6pJ, and $E_{AC}$ (Energy for Accumulate) is 0.9pJ. We will perform an analysis and comparison based on these data. In spiking neural networks, we calculate the number of spike emissions according to the concept of total Synaptic Operations which is named Synops. Unlike ANNs, in spiking neural networks, each spike emission corresponds to an accumulation operation, which results in lower energy consumption per calculation compared to traditional ANNs. Our RSNN network, which we used in previous experiments, takes continuous numerical frames as input to the SNN layer directly after processing by CONVLSTM, effectively serving as an encoder layer. The calculation in the first layer of this encoder is in the form of multiplication and accumulation, similar to CNN operations. This direct encoding of continuous numerical images incurs additional energy consumption. To highlight the energy advantage of SNN, we applied a simplified SNN structure to RSNN and conducted experiments on the DVSGesture dataset. The experimental results also indicate that our RSNN model has a greater energy advantage over the RNN model, with a relatively minor decrease in accuracy consumption. For more specific comparisons, refer to Table 4.

Since the structures of CONVLSTM and convolutional LSTM with attention models are the same as RSNN in the convolutional LSTM part, the difference lies in the subsequent CNN decision and SNN network. For simplicity, the energy consumption comparison is made between CNN and SNN only. In the case of SNN networks, we calculate the number of spike emissions by running one epoch on the test set and then computing the average number of spike emissions for a single image.

## 3 CONCLUSION

In this paper, a dynamic visual cognitive model RSNN based on biological brain is proposed. Inspired by the time memory ability of human visual system neurons for dynamic images, we use the RNN module to simulate the function of this part and take the pulse neural network with the same data transmission mode as biological neurons as the decision-making module. Based on this, we construct the dynamic DVS image model. We show that this model can greatly improve the ability of a single SNN structure network to identify DVS data and achieve better performance with fewer SNN layers through efficient spatio-temporal feature extraction, which can achieve lower energy consumption, and shallower SNNS are convenient for other training methods. In the future, we will further study the recognition decision principle from biological visual systems and apply the relevant structural functions to the performance improvement of SNN model in the field of vision.

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
