# OpenReview forum: "RSNN: Recurrent Spiking Neural Networks for Dynamic Spatial-Temporal Information Processing"
_acmmm.org/ACMMM/2024/Conference — MM2024 Poster_

### Official Review · Reviewer_abaf · 2024-05-19

**Rating:** 5
**Confidence:** 4

**Summary:**

SNNs exhibit benefits in discrete event data processing due to binary computation. However, preprocessing requirements lead to time information loss. We present an RSNN with spiking dynamics to minimize this loss in slices. By incorporating a recurrent structure and CONLSTM, the model outperforms spiking-based alternatives while reducing energy consumption, promising neuromorphic hardware applications.

**Strengths:**

1) The RSNN approach combines spiking neural dynamics with a recurrent structure to minimize time domain loss, which is a novel contribution.
2) The RSNN model is evaluated on event-based datasets and outperforms spiking-based alternatives in terms of accuracy and energy consumption.
3) The paper provides a clear description of the RSNN model, motivation, and experimental results.
4) The RSNN model has potential applications in neuromorphic hardware due to its energy efficiency.

**Limitations:**

1) Ensure that "where" after equations is lowercase.
2) Maintain consistent spacing before references (e.g., on page 224).
3) Add a space after punctuation, such as periods (e.g., on page 308).
4) Consider deleting unnecessary lines (e.g., lines 420-424) to improve conciseness.
5) Separate the methodology and experimental sections to enhance readability.
6) Correct the faint border in Figure 2 for improved visual clarity.
7) Analyze the decrease in the sum of w_att in Table 2 to ensure it does not compromise the validity of the results.

**Suitability:**

3

---

### Official Review · Reviewer_hmUS · 2024-05-23

**Rating:** 5
**Confidence:** 4

**Summary:**

This paper proposes a hybrid neural network architecture that combines Spiking Neural Networks (SNNs) and Recurrent Neural Networks (RNNs) to construct an RSNN model with enhanced spatiotemporal information extraction capabilities. To address the limitation of SNNs in extracting fine-grained texture features from frame sequence images, this paper introduces the mature temporal information processing capabilities of RNNs to improve SNN's recognition ability for video frame sequence data. Moreover, the RSNN hybrid structure retains the feature extraction capabilities of RNNs while inheriting the low energy consumption advantage of SNNs. The paper demonstrates this through energy consumption calculations and comparative experiments.

**Strengths:**

This paper introduces a hybrid neural network architecture that combines RNNs with SNNs . This architecture integrates a pre-trained RNN feature extraction module with an SNN recognition module, preserving the powerful spatiotemporal feature extraction capabilities of RNNs while also maintaining the low-energy advantages of SNNs. Compared to network models with the same structural framework, the RSNN network offers greater energy efficiency with only a minor reduction in recognition accuracy compared to pure RNN networks. Additionally, the RSNN network achieves higher recognition accuracy than SNN networks with the same architecture. This research provides valuable insights for applying SNNs to temporal data processing tasks.

**Limitations:**

The paper has some shortcomings as follows:
1.The pre-training process of the RNN feature extraction module in the paper is not sufficiently clear. Additional sections are needed to describe the specific pre-training process in more detail.
2.There are some spelling and grammar errors in the paper. Some of the illustrations are also rather blurry.

**Suitability:**

3

---

### Official Review · Reviewer_WhdF · 2024-05-24

**Rating:** 2
**Confidence:** 3

**Summary:**

This paper proposed a RNN-SNN hybrid model to learn the event-based spatial-temporal information effectively.

**Strengths:**

1. This paper proposed a RNN-SNN hybrid model, which is a new architecture for the SNN community.

**Limitations:**

1. I tend to think that the theme of this paper is not related to the MM community.
2. Considering that this work did not make some improvement or innovation about the neuron models or learning algorithms of SNNs, I tend to think that its contribution to the SNN community is limited.
3. The experimental results on DVS-Gesture and CIFAR10-DVS are not convincing. Note that previous work [1] has achieved 97.90% and 84.50% on DVS-Gesture and CIFAR10-DVS respectively, which is much higher than the accuracies reported in this paper.

[1] Wang, Z., Jiang, R., Lian, S., Yan, R., & Tang, H. (2023). Adaptive smoothing gradient learning for spiking neural networks. In International Conference on Machine Learning. PMLR.

**Suitability:**

2

---

### Official Review · Reviewer_mUpj · 2024-05-24

**Rating:** 2
**Confidence:** 4

**Summary:**

In this paper, an effective recurrent Spiking neural network is proposed to pre-process the slices using a recurrent structure, which is then further fed into the spike structure to enhance the temporal correlation between the slices.

**Strengths:**

The authors propose a recursive spiking neural network for dynamic spatio-temporal information processing that leverages neural dynamics to achieve optimal performance on event-driven datasets.

**Limitations:**

I believe that this paper has significant weaknesses and as a whole this paper reads like a coursework report rather than an academic research paper, and does not bring anything of value to the research community. I am inclined to reject this paper.

1. The lack of ‘Preliminaries’ and ‘Related Work’ chapters makes it difficult to understand the contribution of the work to the field as a whole.
2. In the description of the methods section, a lot of space is devoted to existing work, such as CNNs and bioinformation processing systems, LSTM network structures, and Attention mechanisms. These contents are simple and should be common knowledge for researchers, and similar extensive presentations should not appear in top conferences such as ACM MM.
3. Experimental comparisons are inadequate, using small network sizes, and comparisons with other related work on RSNN are missing [1].
4. The authors mention in the abstract that ‘traditional data frame segmentation methods lead to the loss of a large amount of temporal information’, so why do the authors still use event frame integration in the methods section? That is, the events are divided equally in the time dimension, and the events within a time period are summed up as a data representation? This approach loses some of the temporal information and is commonly used in spiking neural networks without recurrent network structure.
5. Actually, I think the combination of recurrent and spiking neural networks needs to be carefully considered. It seems to me that the spiking neural network only provides recurrent information through membrane potential leakage, so it is not capable of stronger information like a strong Markov chain like a recurrent neural network. Combining the spiking neural network with a recurrent network structure does bridge the performance gap, but the time cost needs to be considered, in my previous experiments on NLP and ASR in addition to the vision task, the fact that the network requires two layers of loops, the first being the time step of the LSTM itself, and the second being the time step of the SNN, and when the task requires a long time step, the cost of such a cost of nested time steps is good. Most importantly, recurrent SNN network structures do not usually perform better than sequential SNN models.

[1] Liu, Qianhui, et al. "Event-based multimodal spiking neural network with attention mechanism." *ICASSP 2022-2022 IEEE International Conference on Acoustics, Speech and Signal Processing (ICASSP)*. IEEE, 2022.

**Suitability:**

2

---

### Meta-Review · Area_Chair_nYnc · 2024-06-27

**Recommendation:** Accept (Poster)
**Confidence:** 4

**Metareview:**

This paper proposes a hybrid RSNN architecture to extract the time-domain information and reduce computational energy consumption effectively. In addition, an extended version based on CONLSTM has also been explored, which facilitates the applications of neuromorphic hardware. After the discussion period,  one reviewer rates accept, two reviewers rate weak accept and one reviewer rates borderline reject. Overall, this paper may need to be further strengthened in terms of writing style and provide a clearer explanation for the motivation and detailed process of the proposed method. However, it seems to have certain value and inspiration in promoting the development of neuromorphic data processing. After considering the final ratings given by the reviewers comprehensively, the final decision is to accept this paper.